# Big Data Analytics and Machine Learning in Supply Chain 4.0: A Literature Review

Elena Barzizza, Nicolò Biasetton, Riccardo Ceccato  and Luigi Salmaso *

Department of Management Engineering, University of Padova, 35100 Padova, Italy
* Correspondence: luigi.salmaso@unipd.it

**Abstract:** Owing to the development of the technologies of Industry 4.0, recent years have witnessed the emergence of a new concept of supply chain management, namely Supply Chain 4.0 (SC 4.0). Huge investments in information technology have enabled manufacturers to trace the intangible flow of information, but instruments are required to take advantage of the available data sources: big data analytics (BDA) and machine learning (ML) represent important tools for this task. Use of advanced technologies can improve supply chain performances and support reaching strategic goals, but their implementation is challenging in supply chain management. The aim of this study was to understand the main benefits, challenges, and areas of application of BDA and ML in SC 4.0 as well as to understand the BDA and ML techniques most commonly used in the field, with a particular focus on nonparametric techniques. To this end, we carried out a literature review. From our analysis, we identified three main gaps, namely, the need for appropriate analytical tools to manage challenging data configurations; the need for a more reliable link with practice; the need for instruments to select the most suitable BDA or ML techniques. As a solution, we suggest and comment on two viable solutions: nonparametric statistics, and sentiment analysis and clustering.

**Keywords:** Supply Chain 4.0; machine learning; big data analytics; advantages; disadvantages; area of application; nonparametric statistics; sentiment analysis

## 1. Introduction

In recent years, many changes have affected both industry and society, and the relationship between humans and machines [1] has been characterized by higher levels of interaction. The concept of Industry 4.0 first appeared in a 2011 article in reference to a German government program [2] for a 2020 high-tech strategy [3]. The evolution from Industry 1.0 to 4.0 was described in Xu et al. (2018) [4] and Bigliardi et al. (2020) [5]. In summary, the era of Industry 3.0, which started in the middle of the 20th century, is also known as "the information age", which is characterized by the introduction and development of information and communication technologies (ICT), which means manufacturing technologies such as computer numerical control (CNC), computer-aided design (CAD), and so on (see Xu et al. (2018) [4]). The era of Industry 4.0, which started at the end of the 20th century, is also known as "the age of cyber-physical systems (CPSs)". Core developments that characterized this era are represented by CPSs, which allow collaborations between computational and physical entities; in other words, this is an integration between software and components. The term Industry 4.0 refers to the integration of industrial technologies and information and communication technologies [6] to move toward the general automation and digitalization of industrial processes [5]. Industry 4.0 proposes industrial production characterized by strong product individualization and flexibility, greater connection between products and services, and high quality [7]. In particular, Industry 4.0 is linked to benefits such as flexible mass production, cost reduction, creation of new services, introduction of new business models, and real-time coordination and optimization of the entire value chain [5]. Various technologies have a strong relationship with this

fourth industrial revolution: cyber-physical systems, the Internet of Things, big data, cloud computing, 3D printing, additive manufacturing, augmented reality, and autonomous robots [5]. See Bigliardi et al. (2020) [5] for a description of these technologies. Technologies from Industry 4.0 along with significant investments in information technologies and their applications have enabled manufacturers to trace the intangible flow of information. As reported by Alhalalmeh et al. (2022) [8], numerous papers have been published in the field of Industry 4.0 and supply chain management in recent years, and the union of these two concepts has produced the term Supply Chain 4.0 (SC 4.0). SC 4.0 can be defined as "a transformational and holistic approach for supply chain management that utilizes Industry 4.0 disruptive technologies to streamline supply chain processes, activities and relationships to generate significant strategic benefits for all supply chain stakeholders" as reported in the findings of Frederico et al. (2020) [9].

SC 4.0 can be seen as an evolution of SC 3.0. Parallels can be seen between Industry 3.0–SC 3.0/supply chain management 3.0 and Industry 4.0–SC 4.0/supply chain management 4.0. In particular, as mentioned by Frazzon et al. (2019) [7], in supply chain management 3.0, there is an integration of only two channels, while in supply chain management 4.0, the integration is more complicated because it involves the whole supply chain network. Managing an entire network clearly requires more advanced analytical tools and is a challenge compared with the integration of only two channels. As shown below, one of the relevant problems in the implementation of BDA and ML is integration throughout the supply chain. See Frazzon et al. (2019) [7] to better understand the differences between supply chain management 3.0 and supply chain management 4.0 and their core technologies. However, as an example, the technologies used in SC 3.0 include barcodes, enterprise resource planning (ERP) systems, and warehouse management systems (WMS); conversely, as set out by McKinsey [10], SC 4.0 deals with the application of IoT, advanced robotics, and advanced big data analytics in supply chain management. Owing to digitalization, SC 4.0 can achieve velocity, flexibility, granularity, accuracy, and efficiency [10] As mentioned above, the fourth industrial revolution introduced substantial changes: new technologies emerged, and customer expectations have become a key driver for success in the market [10]. As a result, supply chains have had to adapt. Some demographic changes have also influenced the transformation towards SC 4.0: the growth of some rural areas means supply chains must expand geographically, the increasing age of the workforce has led to a need for more ergonomic tools, there are socioeconomic pressures to reduce carbon emissions and pollution, and so on [10]. The International Data Corporation highlighted five key capabilities linked to SC 4.0 (referred to synonymously as the "thinking supply chain") [11], stating it should be connected, collaborative, cyber-aware, cognitively enabled, and comprehensive. In other words, such digital supply chains should have access to as many data sources as possible despite the difficulty of integrating all the data sources; it should improve collaboration with other supply chain actors becaused part of the value creation consistently comes from outside the focal firm; a thinking supply chain should have powerful tools to counter cyber attacks; it should be integrated with a cognitive artificial intelligence platform; and it should have the ability to produce fast, comprehensive analytics in order to support decision making in real time.

Thus, the description of the key capabilities that SC 4.0 should implement highlights the importance of being able to extract useful information from data: the analytical capability is fundamental in such a scenario. What has changed? How has analytics evolved over the years in the field of supply chain management? Analytics was only previously used within the supply chain framework to perform statistical analysis of some important performance indicators or to forecast supply chain demands [12]. In the 1990s, companies began to adopt electronic data interchange (EDI) and enterprise resource planning (ERP) systems to connect and exchange information between supply chain partners. Only around the turn of the century did business intelligence and predictive analytics software start to develop and be used by companies; this implementation led to better knowledge of the entire supply chain and enabled improved decision-making processes and optimization [12].

Today, companies have access to much more data, and tools are required to extract useful information, which represents a challenge [12]. To provide an idea of this increase, in 2017, a company had access to 50 times more data than 5 years previously [11], including both structured and unstructured data. Soon analytics will move toward artificial intelligence solutions to extract information from all these available data [12]. SC 4.0 was discussed and represented (at length) by He et al. (2021) [13]. In particular, SC 4.0 is described as a supply chain in which all data sources are connected, meaning it includes the data generated inside the focal firm (such as purchasing, manufacturing, and distribution data), data from suppliers and customers, and data generated outside the focal firm or its supply chain, such as social network, weather, geopolitical, and other data. Over the past two decades, data generation has increased in all sectors due to information technology: we live in a digital era characterized by the availability of huge amounts of data generated from heterogeneous sources [14]. This is equally true in the supply chain framework, where the use of sensors, smart devices, and IoT solutions helps collect huge amounts of data. The availability of such data represents an opportunity for companies: properly analyzed, valuable information can be obtained to generate a competitive advantage [14]. For this reason, SC 4.0 includes a supply chain big data center where all the data are stored and where a fast, comprehensive analysis can be carried out by applying big data analysis and machine learning techniques. Due to these tools, SC 4.0 can achieve strategic goals and improve general performance [8]. Figure 1 visually represents the above.

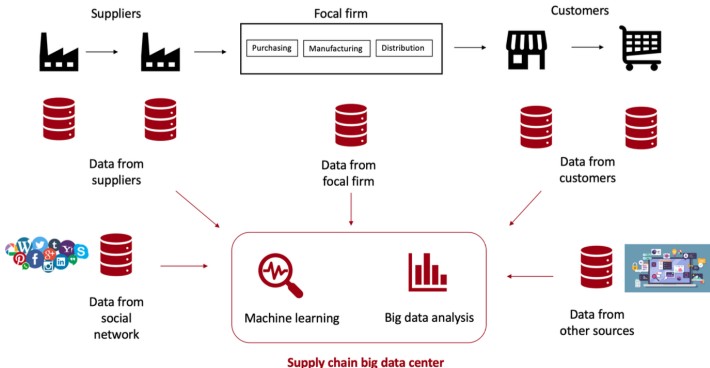

**Figure 1.** Representation of a generic Supply Chain 4.0.

Data represent an important asset for supply chain management. In fact, data can make a supply chain more intelligent, flexible, and adaptive [15]. However, there are some challenges to overcome if we want to introduce a data-driven philosophy into the field of supply chain management, for example, data processing [15]. The problem is that most companies are still in the early stages of adopting big data and machine learning technologies because of a lack of understanding of big data and how to manage it [16]. The aim of this study, therefore, was to focus on the application of such analytical technologies in SC 4.0.

Big data is defined as "the collection and interpretation of massive data sets, made possible by vast computing power that monitors a variety of digital streams" (Davenport (2014) [17]). According to Gartner [18], big data is an informational asset with some peculiar characteristics, namely, volume, velocity, variety, veracity, and value. In more detail, *volume* refers to the amount of data, *velocity* is the speed at which data are generated (here, we deal with real-time data generation), *variety* is the heterogeneous sources of big data, *veracity* is the errors and noise in data, and *value* is the potential for creating a competitive advantage by analyzing these data. The means by which it is possible to analyze big data is known as big data analytics, and this can be defined as the "collection and interpretation of massive data sets, made possible by vast computing power that monitors a variety of digital streams—such as sensors, marketplace interactions and social information exchanges—and analyses them using 'smart' algorithms" (Barbosa et al. (2018) [19]). Big data analytics

makes reference to three analytical methods: descriptive, predictive, and prescriptive analytics. More specifically, *descriptive analytics* refers to the use of tools to describe the current state of a situation by responding to the question regarding what is happening; *predictive analytics* implies the use of algorithms to predict future events based on historical data by answering the question regarding what might happen in the future, searching for correlations in data to generate a forecast for the future; *prescriptive analytics* concerns the use of algorithms for optimization purposes in order to move in a more desirable direction by discovering the causal relationships in data.

The big promise offered by big data analytics concerns machine learning. Shaharabi et al. (2009) [20] defined machine learning as "a scientific algorithm developed in order to recognize the patterns to forecast the estimates of activity". In other words, a machine learning algorithm continuously learns from data inputs and creates its own rules and models by recognizing the patterns within data. There are three types of machine learning algorithms, namely, supervised, unsupervised, and reinforcement learning. In *supervised learning*, the algorithm receives an expected output as an input and uses it to create the rules and models. In *unsupervised learning*, the output is not known, so the algorithm has to recognize patterns within the data. Finally, in *reinforcement learning*, the desired output is not known but the algorithm interacts with the external environment to understand if an action is correct or not.

Defining these two analytical tools naturally raises the question: why is studying big data analytics and machine learning in SC 4.0 of interest? The literature reveals different opinions on opportunities offered by the application of big data analytics in supply chain management, and the debate is vast [21]. In this era of modern technology, obtaining valuable insights from data is considered to be a key issue [22]. Owing to the application of big data analysis and machine learning algorithms, it is possible to derive an advantage from the data sources available to a company. Indeed, some authors argued that the application of such tools can help supply chains create new competitive advantages and improve operational effectiveness [23]. Despite the enormous opportunities derived from the application of big data analysis in the supply chain field, its implementation also faces major challenges [15]. One of the biggest challenges to its adoption in manufacturing contexts is the lack of skills and appropriate techniques [24]. Similarly, authors recognized that the practical implementation of the techniques related to Industry 4.0, including big data and data mining, is a barrier that needs to be overcome [7] if companies want to fully exploit the benefits of these tools. This is confirmed by the fact that companies are not always able to obtain the desired benefits from investments in analytical tools despite the importance of data processing to optimize decision-making processes. In short, in the literature, scholars and practitioners recognize the potential of big data in supply chains but highlight the existence of barriers that must be overcome if companies wish to obtain positive returns on this type of investment.

It follows, therefore, that an investigation of some important aspects related to the application of big data analytics and machine learning in the field of SC 4.0 is needed. Specifically, an in-depth understanding of the benefits obtained from the application of these technologies in the SC 4.0 environment and what barriers still exist is required. This led to two study questions:

*LRQ 1* What are the advantages of using big data analytics and machine learning in SC 4.0?

*LRQ 2* What are the challenges with using big data analytics and machine learning in SC 4.0?

Another aspect of interest is the areas benefited within SC 4.0 by big data analysis and machine learning. This consideration led to another study question:

*LRQ 3* In which areas of SC 4.0 is there the greatest need to apply big data analysis and machine learning techniques?

Finally, our literature review had another more technical aim, which was to understand which big data analysis and machine learning techniques are mentioned by the authors in

literature in the field of SC 4.0. In particular, it is recognized that real data do not always follow a normal distribution [25]; therefore, the application of a nonparametric statistic could be a viable solution to perform analyses on such data. The last study question was therefore:

*LRQ 4*   What machine learning or big data analysis techniques are used within SC 4.0? Are there any references to nonparametric methods?

In this study, we aimed to cover these research questions in order to provide structured clarification of the comprehensive knowledge of the use of BDA and ML within the new framework of SC 4.0. We think that it is important to understand the role of statistics in our society and, in particular, its role in today's industry context. Indeed, understanding the needs of society and industry may help us develop appropriate and viable analytical tools, and only this will allow us to make relevant contributions to scientific literature as a whole.

The paper is organized as follows: in Section 2, we describe the methods applied to answer the four research questions, i.e., the keyword query, the selection funnel, and the description of the dimensions of interest; in Section 3, we explain the results of the literature review; in Section 4, we discuss some possible solutions to deal with the typical data generated in SC 4.0.; in Section 5, we present some final remarks.

## 2. Materials and Methods

To answer the four research questions presented in Section 1, we performed a literature review using two databases, namely Scopus and Web of Science. The performed query (see Figure 2) was split into two main blocks, one concerning the concept of SC 4.0 and its associated terms and the other concerning big data and machine learning.

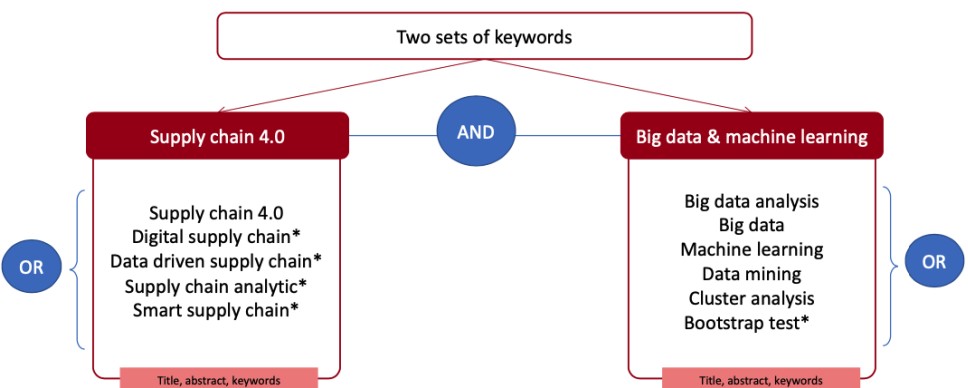

**Figure 2.** Query performed both in Scopus and Web of Science.

The first set of keywords contained words "supply chain 4.0", "digital supply chain*", "data-driven supply chain*", "supply chain analytic*" and "smart supply chain*", all linked by the logic operator *OR* Similarly, the second set of keywords containing "big data analysis", "big data", "machine learning", "data mining", "cluster analysis", and "bootstrap test*" was also linked by the logic operator *OR*, and the two blocks were linked by the logic operator *AND*. Both sets of keywords were searched for in the titles, abstracts, and keywords of both databases.

The query was performed on 15 November 2022, and we found 157 papers in Scopus and 160 in Web of Science. A filter was applied to exclude articles not in English. The two sets of results were then merged and duplicates removed, leaving 231 papers. Each abstract was read to identify the papers that aligned with the literature research aims, leaving 104. These were then read in full, and in the end, we had a subsample of 66 articles for our literature analysis. Some articles were excluded because they not align with the aforementioned research questions. Figure 3 displays the selection funnel.

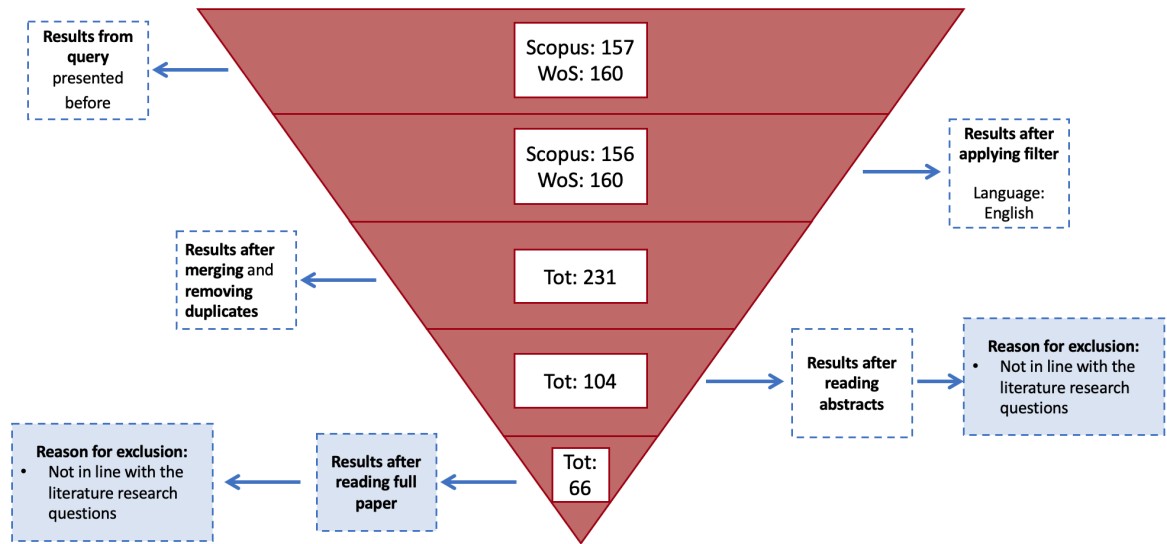

**Figure 3.** Depiction of the selection funnel.

One aspect worth considering is the trends in the publication of these papers. Figure 4 represents the publications over time of the 231 papers obtained after merging the results of the two databases (Scopus and Web of Science) and removing duplicates. The first paper appeared in 2009, but the field of research has been developing since 2016. The increase in publications over the last decade, bar 2022, may signify that interest in this field of research is growing year-by-year. The drop in 2022 can be attributed to the COVID pandemic. Indeed, COVID and its related uncertainties hindering the implementation of big data and machine learning projects is confirmed by a 2021 [26] survey in which it was recognized that uncertainties linked to COVID represented one of the main barriers to digitalization.

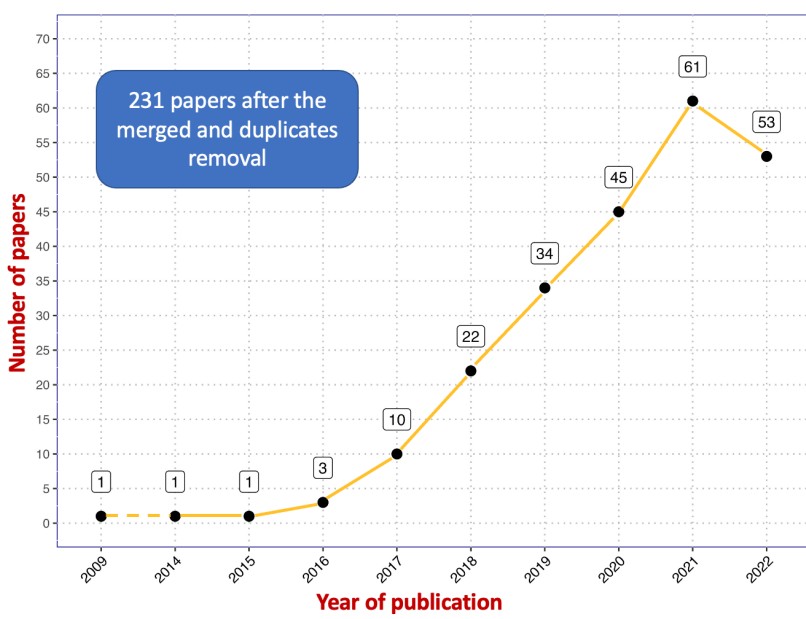

**Figure 4.** The trend in the publication of the 231 papers obtained after merging the two databases and removing duplicates.

We examined the final 66 papers involved taking note of the following details of interest:

- Title : title of the publication.
- First author: first author of the publication.
- List of authors: all authors who contributed to the publication.

- Year of publication: the year in which the paper was published.
- Type of contribution: whether theoretical or practical, whether focused on a theoretical or practical issue.
- Research methodology: the methodology used to achieve the research aim, such as literature review, case study, or survey.
- Positive impact of big data analytics and machine learning: we considered the positive aspects mentioned by the authors related to the application of big data analytics and machine learning in SC 4.0.
- Challenges of big data analytics and machine learning: we considered the challenges and barriers mentioned by the authors related to the application of big data analytics and machine learning in SC 4.0.
- Area of application: we collected information about the fields of SC 4.0 in which big data analytics and machine learning have been successfully applied.
- Data sources: type of data sources, i.e., internal, external, or both.
- Data type: types of data commonly dealt with in SC 4.0, e.g., structured, semistructured, or unstructured.
- Data distribution: we collected information about the distribution of data typically found in SC 4.0 (normal, non-normal, unknown, etc.).
- Big data analysis techniques: we noted which big data analysis techniques are most commonly found in SC 4.0.
- Machine learning techniques: we noted which machine learning techniques are most commonly found in SC 4.0.
- Non-parametric methods: we noted which nonparametric methods, if any, are applied in SC 4.0.

## 3. Results

### 3.1. Descriptive Statistics

This section looks at some descriptive statistics in relation to the selected sample of 66 papers (see Figure 3).

The papers were firstly split into two classes (see Figure 5), i.e., those focusing on a theoretical aspect of big data analytics and/or machine learning in SC 4.0, and those focusing on their practical application. Most of the articles in the literature (85%) provide a theoretical discussion about the implementation of big data analytics and/or machine learning in SC 4.0, with only a small percentage (15%) focused on a practical application in this context.

Type of contribution

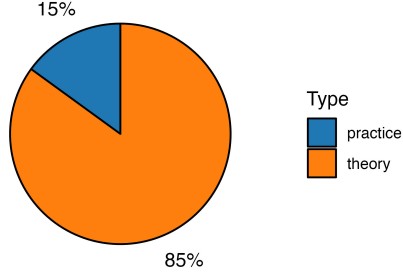

**Figure 5.** Classification of studies based on a theoretical or practical focus.

To better understand the general characteristics of the sample of papers, the employed methodology was also considered (see Figure 6). Most researchers used a literature review (44%) to answer their specific study questions, followed by research papers (27%), then some proceedings papers (14%), and finally some case studies (15%) developed in this field of research.

Methodologies applied

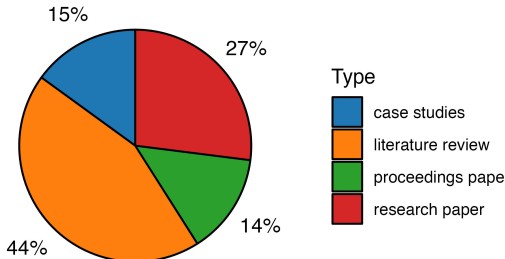

**Figure 6.** Types of methodologies applied in the subsample of studies considered.

Another important aspect to consider concerned the data. As stated in the Introduction, SC 4.0 is a supply chain in which all the data are connected. This means we have data from different sources, both internal and external to the focal firm and, in general, to the supply chain. Of the articles, 65% mentioned data generated from inside the company, while 32% referred to data from the external and internal environments; in 3% of cases, the authors mentioned data derived from external sources (Figure 7).

Data sources

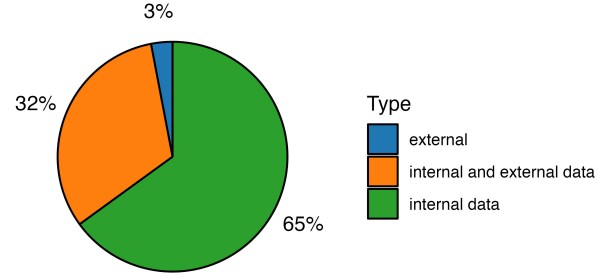

**Figure 7.** Types of data sources.

A further aspect of great interest is the type of data circulating in SC 4.0 and available in the supply chain's big data center. Owing to information technologies, manufacturers can trace the intangible flow of information characterized by data that are structured, unstructured, or both. By structured data, we mean traditional data that can be represented using tables and can thus be analyzed using traditional statistical tools. Alternatively, owing to techniques such as webscraping, it is possible to collect data generated by social networks, such as comments, images, videos, recordings, and so on. These types of data are unstructured and require specific analysis tools involving sophisticated machine learning algorithms if we wish to extract useful insights from them (i.e., sentiment analysis). As far as the papers in this literature review are concerned, 29% of them focused on structured data, 13% on unstructured data, and the majority, 58%, on both structured and unstructured data (see Figure 8). This result highlights a significant need for appropriate analytical tools that differ from traditional tools.

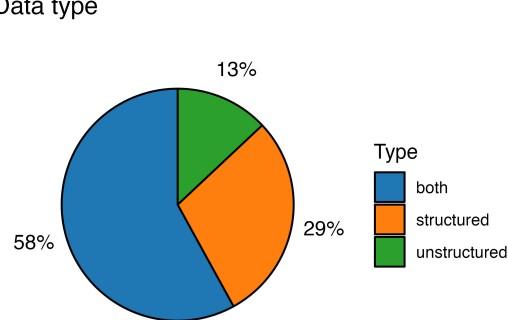

**Figure 8.** Types of data.

*3.2. Analysis of the Papers*

Analysis of the papers throws up several observations. The concept of SC 4.0 was only introduced in the past decade and many authors have focused on studying the positive impact of big data analytics and machine learning in this context.

One of the most-cited advantages of using big data analytics and machine learning is the improvement in all types of performance of a company from different perspectives; of particular note, operational and organizational performance is enhanced ([27,28]). Owing to digitalization, companies have the possibility to improve the performance of flexibility in the entire supply chain [29]; as stated by many authors in the literature (i.e., [16,30–35]), the application of big data analytics in supply chain management can enhance the operational efficiency of a company. These analytical tools also have a positive impact on innovation performance [36]. The use of analytics can help a company improve the allocation of resources [37]; better integration is yet another example of the benefit of big data analytics in SC 4.0 [38]. A further significant improvement concerns production performance: a company's production system is optimized [39] and annual productivity shows an overall improvement [37]. In Liu et al. (2021) [40], production efficiency showed an increase of 30%. With analytics, it is possible to adapt productivity in terms of speed, mix, or volumes [41] due to better production scheduling [42]. Delivery is also positively impacted with, for example, product delivery being more accurate [41]. Cost performance is also enhanced due to the use of analytics in SC 4.0 [43], for example, inventory costs due to a better forecast process [44] or operation and communication costs [8]. Handanga et al. (2021) [42] reported that big data analytics can reduce costs by 20–50%.

The reduction in costs is also linked to a reduction in inventory, in particular to safety stock [45] and inventory levels [29]. In particular, data analytics allows a reduction in batch size [46].

In addition to a reduction in costs, these analytical tools enable a significant increase in the financial performance of the entire supply chain. Handanga et al. (2021) [42] reported that big data analytics leads to an improvement in profit margins of around 2–3%, and the revenue of a company might increase due to more accurate product fraud detection by using appropriate machine learning algorithms [47]. Generally, the availability of better forecasting tools, which integrate machine learning models, rather than traditional ones, such as those used by retailers (see Sardar et al. (2021) [44] for a detailed description), allows the profitability of a company to increase [44]. In general, supply chain analytics shows a high return on investment [37] (about 15–20% [24]).

Another benefit to consider is the possibility of minimizing risk [48].

A well-cited positive impact of implementing big data analytics and machine learning is the reduction in time. The application of these tools ensures a reduction in lead time [43], time to market [32], development time [49], and reaction time [13], which allows for a quick response to market changes [50] and, in general, a faster responsiveness [51].

A further positive aspect concerns quality [46]. One of the strategic benefits is improvements in quality for both products and services [8].

Several informational benefits have emerged from the application of analytics in SC 4.0. The analysis of data allows for more targeted decisions [52] and improves the entire process of decision making [53]. In fact, analytical tools allow easier and quicker access to data [8]. Other informational benefits include information exchange [16], information processing [54], information sharing [38], extrapolation of strategic information from data by discovering insights [35], use of knowledge not previously considered [55], timeliness of data processing [56], and reductions in uncertainties [36].

Informational benefits also have a positive impact on the discovery of new business opportunities, thus creating a competitive advantage. One of the strategic benefits derived from implementing big data analytics in SC 4.0 is the construction of a competitive advantage for the company [8], for example, by discovering new business opportunities [8] or developing potential markets [35]. The fact that using big data analytics makes it possible to respond to consumer needs with an appropriate product/service creates a competitive advantage for the company [45], especially for those operating in a dynamic market.

Benefits are seen not only in a single factory but also in the whole supply chain. There is a general improvement in supply chain performance such as better interconnectivity between actors in the same supply chain [56] and better relationships between them [32] due to quicker responses [16], transparency and visibility [54], agility [32] and sustainability [27]. Regarding this last point, big data analytics makes it possible to build a green supply chain [24]. The positive effects of an improvement in supply chain relationships can also be seen in supplier management [34]. Analytics supports the supply chain in the creation of new value [54] and, especially, in the creation of new customer value [57].

Another important benefit recognized alongside value creation is the improvements in customer satisfaction and customer relationships. Indeed, big data analytics and machine learning make it possible to move toward a more customer-oriented approach [46] and create additional value for customers [58]. Due to improvements in costs, quality, flexibility, and delivery, it is possible to increase overall customer satisfaction [38] by cultivating relationships with customers [8]. Additionally, owing to analytics, it is possible to facilitate an understanding of customer needs [37] and enhance consumer loyalty [59]. Big data alongside social media enables companies to engage with customers during the entire process of creating new products or services [31]. Finally, as previously explained, better forecasts of customer demand through the use of appropriate algorithms makes it possible to introduce new business opportunities [31].

Generally, all functions of a company and the entire supply chain are positively affected by a better forecasting process that provides more robust demand or sales forecasts [60] and less errors in the whole purchasing management process [61]. This has an impact, for example, on both service level and inventory costs [62]. A better demand forecast process positively impacts marketing activities [13]. In particular, when dealing with time series, data machine learning algorithms show higher predictive performance than traditional methods [63].

Finally, analytical tools facilitate the overall optimization of products or services of an already existing or new company; the optimization of internal processes, including decision processes [55], such as the assembly process [37]; as well as the optimization of production lines, allowing a reduction in defects [40].

To sum up, our analysis shows that the advantages can be grouped into two main classes: performance advantages (operational, organizational, innovation, and financial benefits) and informational advantages (interconnectivity, visibility, transparency, accurate forecasting, and targeted decision processes). Therefore, expanding the use of these analytical tools in this context brings important advantages to companies and the whole industrial context.

Despite the numerous benefits of applying big data analytics and machine learning in SC 4.0, the challenges and barriers to their implementation cannot be ignored and deserve special attention in order to find appropriate solutions.

One of the main barriers concerns data and data processing. It is necessary to choose the right data strategy, collect the required data, properly share the data, and find the appropriate algorithm to analyze the data [25]. These are challenging tasks because the literature provides no clear guidance for companies on how to make use of digital data [25]. In other words, there is no roadmap to guide the process of analyzing data [64]. A more technical problem concerns data storage and transmission [65]. With regard to the data, authors have recognized problems with data quality and scalability due to the huge quantities of data [42] as well as data accuracy and diversity [66]. Due to the particular characteristics of big data mentioned in Section 1, their management is challenging [46]; nowadays, only a small percentage of data are analyzed and exploited to enhance decision-making processes [15]. The problem is therefore how to extract commercial value from the data flowing in SC 4.0 [13], particularly as we are dealing with data in different formats [27]. Data complexity is not negligible; therefore, the analysis represents a challenge [67]. Additionally, a trade-off must be found between accuracy in analysis, computing resources and time required [63], and the implementation of advanced methods [30].

It is particularly challenging for practitioners to create a link between research and practice [55] in order to create real value for the supply chain. In the literature, there are solutions that develop methods that are not suitable for application [65]. This is further confirmed by the low number of case studies in the literature focused on actual implementation within the SC 4.0 framework [68].

The lack of empirical examples is probably due to a lack of mathematical knowledge and lack of professionals specialized in this field [37] as well as talent in big data analysis [69]. Indeed, in order to fully exploit the potential of big data analytics, adequate competences are required [19], but there is a lack of knowledge of what big data is and how it can help achieve better performance [36]. There is a lack of capacity necessary to support the company in developing its potential [70].

In industry, the process required to actually implement big data analytics within a company is complex. Indeed, it requires dealing with intangible resources such as information [43]. Being complex, its implementation requires long development times [59] and is time-consuming [42]. The cost of implementation is another factor that cannot be ignored, particularly the technical setup costs [59] and the cost of the infrastucture needed for the implementation of big data analysis [71]. In general, big data technology incurs huge costs [50], and the returns on investments of these projects are not always clear [42]. Implementation is made even more complex by the inadequacy of IT infrastructure [45], together with the fact that there is the need to focus on the entire supply chain [38]. Furthermore, all coordination issues involving all supply chain partners must be overcome [45] as well as integration issues [72], which are particularly evident in global supply chains. Thus, the complexity of the process requires support from senior management, but authors agree there is a lack of movement in this direction [40].

Several operational issues must also be addressed. Firstly, transparency and visibility along the entire supply chain must consider security problems [73], data ownership, and privacy [38]. The implementation of a big data system leads to ethical issues [37], and there are behavioral problems associated with the need to adapt to changes caused by the implementation [42]. Finally, there is a general fear of losing confidential data [31].

Although BDA and ML show important performance and informational benefits, authors still agree that there are substantial barriers to the implementation of these technologies. Firstly, there are problems associated with the complexity of implementation of BDA and ML tools in SC 4.0: the huge costs of implementation, the need for senior management support, and the inadequacy of IT infrastructure. However, these problems are not insurmountable, especially if companies fully understand the enormous advantages linked to BDA and ML implementation in SC 4.0. A very relevant problem is the challenge

of effectively using data and data processing. There is a need to understand how companies can manage data profitably: only by having this understanding can companies be encouraged to invest in this type of project. As such, the literature needs to provide feasible solutions for companies. Below, we discuss some possible directions for development of statistical tools to deal with typical SC 4.0 data.

From the analysis of the advantages and disadvantages of applying big data analysis and machine learning techniques in SC 4.0, we already understand that the tools can be applied in different areas of a company or to support the whole supply chain. One of their most important areas of application, mentioned in 55% of the papers, relates to logistic and inventory management, which includes transportation and distribution. The tools have been widely applied to make forecasts so can therefore be applied to support all functions of a company and the whole supply chain. Over half of the papers (55%) cite cross-functional applicability. The third area of application in order of importance is procurement, mentioned by 48% of papers: big data analytics and machine learning can help with the process of selecting the most appropriate suppliers through a partner evaluation analysis, in supplier relationship management, and in purchasing. Production (42%) is another field in which several advantages can be gained, for example, for production planning activities, customization, fraud detection, inspection and control of materials, maintenance, and by supporting all the company's manufacturing functions. Of the papers, 17% reported that analytics is applied in research and development, particularly in the process of product development and design. Another area of application is sales, mentioned by 11% of the authors, marketing (9%), and customer relationship management, which includes customer service (9%).

To deal with the last study question, we looked at the machine learning techniques cited by the authors. In the 66 papers considered, we found examples referring to all three categories mentioned in the Section 1, in particular, algorithms for both regression and classification, i.e., regression and decision trees, support vector machines, random forest, neural network, K-nearest neighbors, logistic regression, XGBoost, and so on. As with the big data analysis techniques, almost all authors who cited such analytical tools mentioned examples of predictive and descriptive analytics. Only two articles briefly mentioned nonparametric methods, namely, Cavalcante et al. (2019) [25] and Manjunath et al. (2018) [74].

Lastly, another aspect of interest is the fact that, within the analyzed papers, 35% mentioned the use of external data in analytics for SC 4.0. The importance of using external data has been highlighted by several authors. As stated in Thekkoote et al. (2022) [38], a company can achieve very good operational performance by using big data collected from both inside and outside the company. External data can be used as the input for machine learning models in order to, for example, forecast intermittent demand. There may be insufficient internal data; therefore, the use of external data can be a viable solution [62]. Indeed, better forecasts lead to lower inventory costs and higher service levels. Another source of external data is consumer data, which could generate useful insights for the creation of new products [37]. Additionally, consumer data can help both R&D and production by reducing reaction times and costs in the supply chain [13]. The positive impact of using data from different sources can also be seen in logistics [67]. In general, the use of data from different sources is fundamental for the decision-making processes [60].

## 4. Discussion

### 4.1. Knowledge Gaps

From the analysis of the selected papers, some knowledge gaps emerged.

One of the main issues recognized in the literature concerns the management of challenging data configurations. In the manufacturing field, in particular, data are commonly imbalanced [68]. For example, the number of defective products in a production line is greatly inferior to the number of products with no defects, and these data imbalances can cause problems in the application of machine learning algorithms. Moreover, the authors agree that real data do not follow a normal distribution [25] or, in general, a

known distribution of data [44]. As the assumption of normal distribution is violated, it is not possible to apply the traditional tools from parametric statistics. Despite the recognition in the literature that data do not follow a normal distribution, only two articles (Cavalccante et al. (2019) [25] and Manjunath et al. (2018) [74]) briefly mentioned the nonparametric statistic, which would be a viable solution in this case. Another concern regards the dimensionality of data. Brintrup et al. (2020) [68] stated that this is an issue that affects statistical analysis. High-dimensional or "thick" data is a term that refers to a configuration in which the number of considered variables is higher than the number of statistical units; this is common in any industrial application field. Big data therefore refers to large numbers of observed variables. Our analysis of the papers showed that there is a need to understand which big data analysis and machine learning techniques can be successfully applied to deal with challenging data configurations (imbalanced, thick, and non-normal data). If a nonparametric statistic is to be a viable solution, it is necessary to demonstrate that it represents a viable tool for analysis within SC 4.0.

In our article review, we identified a second gap. The literature needs to link in more with practice [55]. Only a few empirical examples linked to the application of technologies from Industry 4.0 [37] included big data and machine learning. Additionally, there is a lack of case studies that demonstrate affordable applications of big data analytics in supply chain management [32]. The practical application of big data analytics and machine learning is seen as one of the biggest challenges in supply chain management [7]. To sum up, the need for focus on the practical applications of big data analysis and machine learning techniques to demonstrate the real potential for value and benefit creation in a manufacturing context is thoroughly highlighted in the literature and confirmed by the review conducted in this study. As shown in Figure 5, almost all publications linked to big data analytics and machine learning in SC 4.0 have analyzed a theoretical aspect, while only 15% have dealt with a real application. As such, a practical demonstration of the applicability and potential of these analytical technologies is urgently required.

In this direction, some relevant works have recently been published in the literature. The examples are all related to strategies for optimization, particularly of the production–inventory system. They discuss mathematical models involving variables from different sources to maximize profit. Mishra et al. (2020) [75] considered the situation characterized by the need to limit carbon emissions; Bachar et al. (2022) [76] investigated the use of partial outsourcing to satisfy customer demands; Padiyar et al. (2022) [77] considered problems of product deterioration and inflation that introduce uncertainties into inventory management, and considered variables from suppliers, producers, and retailers. Sarkar et al. (2022) [78] focused on the use of emerging technology in supply chain management, namely, radio frequency identification (RFID), to optimize warehouse layout. We leave to the reader the further perusal of these examples. All contributions mentioned above are supported by numerical examples to prove the feasibility and convenience of the presented models. However, as stated in Section 3.2, BDA and ML can be successfully applied in numerous areas; therefore, more effort should be made to demonstrate in practice the applicability and convenience of using such tools in all contexts of SC 4.0.

Finally, Awanu et al. (2021) [79] identified a need for suitable tools to help choose the most suitable big data or machine learning tools. In terms of choice of machine learning algorithms, practitioners have usually evaluated the performance of a machine learning model based on certain performance measures (i.e., mean squared error, root squared error, and so on), but there might be an instrument that makes it possible to select the best machine learning model in a robust way. The possibility of having such tools could help practitioners maximize the performance of an algorithm.

We discussed the main advantages of applying big data analysis and machine learning techniques in SC 4.0 as well as the main challenges encountered by practitioners. This field of research has been developing over the past decade, and there are still some knowledge gaps such as those detailed above. Following on from what was said previously about the types of data typically encountered in SC 4.0 (both structured and unstruc-

tured, high-dimensional, and non-normal data), in the following paragraphs, we suggest some statistical methodologies that are suitable for this application context, namely, the nonparametric statistic, sentiment analysis, and clustering.

*4.2. Nonparametric Statistics*

The analysis of high-dimensional multivariate data can be quite challenging, especially when traditional distributional assumptions, such as multivariate normality, do not hold. High-dimensional data involve the presence of a large number of variables, from which we want to extract as many useful insights as possible. These data can be noisy and contain multiple data types, e.g., continuous, binary, ordinal, or even mixed types. Nonparametric techniques are therefore fundamental for the analysis of such data because they do not require strict assumptions on data distribution. Additionally, the observed variables can potentially present high correlation, an aspect that we need to take into consideration when choosing a statistical method for the analysis of high-dimensional data.

Among the nonparametric techniques, multivariate-permutation-based tests [80–82] can be particularly suitable for the analysis of high-dimensional data, because they do not require strict assumptions on data distribution, sample size, or number of variables. In particular, nonparametric combination (NPC) is a permutation-based methodology that has been widely adopted in the context of multivariate problems with a large number of features. This methodology requires only mild assumptions on data distribution, i.e., exchangeability between groups, and can be easily applied to complex problems, such as scenarios involving multiple data types, stratification, or the comparison of multiple populations. Let us therefore focus on this flexible approach.

After defining the system of hypotheses for the problem we want to investigate, this system is decomposed into as many subsystems as the number of observed variables, and, for each of them, we apply a test statistic and use a permutation approach to estimate a *p*-value. The choice of test statistics is unrestricted and is simply driven by the nature of the data (e.g., binary, ordinal, or continuous) and of the comparison we want to conduct (e.g., if we are interested in comparing populations in terms of their cumulative distribution functions, mean values, or standard deviations). Different test statistics can be used at the same time too [83]. This makes NPC-based tests particularly flexible and suitable for a large variety of scenarios. It is worth noting that the correlation between the variables here is taken into account by simply applying the same permutation scheme to each subproblem, which means that we permute rows of our dataset instead of independently reassigning observations of different variables to groups.

The achieved partial *p*-values are then combined by means of appropriate combining functions, so that the original multivariate system of hypotheses can be solved using the resulting combined global *p*-value. The adoption of an appropriate combining function is a fundamental step of NPC, which is again driven by the nature of the problem [81].

Such an approach is particularly suitable for the analysis of high-dimensional scenarios because of a specific property derived from use of the aforementioned combining functions, called the finite-sample consistency [82]. As the number of observed informative variables increases, the power of NPC-based tests increases too. Therefore, when a large number of variables is gathered, these tests are commonly able to easily detect departures from the null hypothesis when they exist.

For the sake of completeness, we searched the literature for other solutions that combine nonparametric statistics with the concept of SC 4.0. We performed a search on Scopus using the first set of keywords in Figure 2 and "non parametric" OR "nonparametric". We found only one article (Yildiz et al. (2022) [84]) in which the authors proposed a methodology that aims to monitor distribution by using a nonparametric control chart. This methodology does not require distributional assumptions. Based on the advantages mentioned above, it is clear that there is a need to thoroughly investigate the use of nonparametric statistics in the SC 4.0 framework.

### 4.3. Sentiment Analysis and Clustering

In an integrated SC, looking at outside the focal firm, ML can help segment the market, identifying groups of customers with similar characteristics. Cluster analysis is among the most-adopted methodologies to separate subjects into groups and works in a way that each subject is more similar to those belonging to the same group than to those in different groups. The identification of market segments can be used by companies to target different groups (with different characteristics, needs, and behaviors) with different strategies, thus improving both profitability and customer satisfaction. Most of the cluster analyses conducted in the customer satisfaction field are based on information regarding motivations, satisfactions, opinions, and judgements. The most common way to measure such constructs is still the adoption of survey questionnaires and Likert-type scales. Multiple procedures have been proposed to deal with the characteristics of Likert-type scales in cluster analysis (see Biasetton et al. (2023) [85] for more details).

The survey approach is labor- and material-intensive and time-consuming and does not obtain large amounts of information. Furthermore, it suffers from selectivity bias because the questions asked stem from practice and the researcher's perspective, thus limiting insight into customers' views. As a starting point to overcome such problems, it can be observed that with the advent of big data and Web 2.0, the availability of data from outside the focal firm has risen, and unstructured data from websites, social media, etc., have proven to be of great value in understanding customer satisfaction, improving profitability, and managing the SC, adapting the offer to customer needs, behaviors, and opinions. In fact, companies and brands are changing the way they engage with their audience as well as the way they maintain the customer relationship. In this context, great importance is given in the literature to textual data; posts written on social media, blogs, discussion forums, review websites, news websites, etc. allow people to express their opinion on products and services as well as their needs. Thus, textual data have been established as a way to measure the customer satisfaction voluntarily expressed by customers, with no external stimuli or cost to researchers given that reviews are easily gathered in real time. Given the unstructured nature of such data, techniques are needed to extract value from them: natural language processing (NLP) refers to the branch of computer science—and of artificial intelligence—concerned with automatically understanding text and spoken words. Text mining, in particular, in its general definition concerns the extraction of structured and high-quality information from textual data. It usually involves the process of structuring the input text into databases and deriving patterns through statistical analysis and, finally, evaluating and interpreting the output.

Of the various text mining techniques, sentiment analysis (SA)—also referred to as opinion mining—can be useful for extracting value and helping to develop new solutions to attract customers. SA is the process of computationally identifying and categorizing opinions expressed in a piece of text, with particular interest in determining whether the writer's attitude toward a particular item (i.e., topic, product or service) is positive, neutral, or negative. Over the past 15 years, different methodologies, which can be grouped into 3 main approaches, have been proposed in the literature to conduct SA [86]: (1) ML-based sentiment analysis; (2) lexicon-based sentiment analysis; (3) hybrid approaches.

Lexicon-based SA is an unsupervised method that works by exploiting lexical resources called sentiment lexicon, or opinion lexicon, which is a predefined list of sentiment words associated with their semantic orientation by a positive or negative score [87,88]. Many lexicons have been developed over the years, some of which also make emotion analysis (EA) possible. EA represents an extension of SA, making it possible to identify emotions and not only the polarity of a text. EA makes it possible to obtain an in-depth description of consumers' underlying feelings and emotions expressed in the text using a much more complex system of analysis. Examples of emotion that can be extracted using the NRC word-emotion association lexicon (aka EmoLex) were defined by Plutchik [89,90] as anger, fear, anticipation, trust, surprise, sadness, joy, and disgust.

A new trend investigates customer satisfaction using cluster analysis data containing both textual and Likert-type ratings (in many situations, available at the same time on websites, review platforms, etc.). The main advantages of this methodology lie in the use of huge amounts of data, beyond the reach of classic surveys, collected from the spontaneous collaboration of users, and in combination with rating analysis with useful information obtained from text reviews.

From the performed literature review, an increasing number of publications were observed in the field of NLP applied to supply chain management. Some applications of NLP to supply chain management include (but are not limited to) the automation of manual processes such as reading shipment documents to provide information to improve logistics, monitoring real-time changes in internal data, gathering benchmark data to exploit new opportunities, reducing language barriers, and using web scraping and social media listening to analyze data and gain business intelligence. NLP enables supply chain organizations to gather and monitor external data that can identify potential disruptions. Supply chains can build resilience to handle crises and mitigate risks with suppliers, manufacturers, and other supply chain stakeholders by taking preventive measures through analyzing reports, industry news, social media, and other areas.

Twitter, for example, has emerged as one of the most widely used data sources for research in academia and practical applications (Singh et al., 2018 [91]). In the literature, there are various examples associated with practical applications of Twitter information to supply chain management, such as brand management (Malhotra et al., 2012 [92]), stock forecasting (Arias et al., 2014 [93]), and crisis management. It is anticipated that there will be a swift expansion in the utilization of Twitter information for numerous other purposes, such as market prediction, public safety, and humanitarian relief and assistance.

## 5. Conclusions

The application of technologies from Industry 4.0 in supply chain management creates what we refer to as SC 4.0. In SC 4.0, different data sources are connected, providing access to huge amounts of data flowing in real time in the supply chain. To gain an advantage from all the available data sources, appropriate analytical tools must be applied. SC 4.0 includes a supply chain big data center [13], where it is possible to apply big data analysis and machine learning techniques to perform comprehensive and fast analytics that support all decision-making processes in the entire supply chain. Indeed, obtaining valuable insights from data is considered key in this era of modern technology [22].

The literature recognizes that the application of such analytical tools represents an important opportunity for supply chains but also poses some considerable challenges [15]. Therefore, to understand the advantages and disadvantages of using big data analysis and machine learning techniques in SC 4.0, what areas of application of such technologies exist, and what techniques are mentioned, we performed a literature review using Scopus and Web of Science.

This field of research has been developing since 2016, and the last decade has seen an increase in the number of papers published, confirming the topic is of interest.

The considered articles show the potential benefit of BDA and ML in two main directions: performance and informational benefits. The problems associated with the complexity of implementing BDA and ML in SC 4.0 cannot be ignored, but the real problem is the need for appropriate analytical techniques to properly analyze and process data. To this end, statisticians can work to provide the analytical tools required for the current industrial context.

As previously stated, our analysis identified several knowledge gaps. Firstly, there are issues with the management and analysis of the data that generally flow in an SC 4.0. In particular, manufacturing data are very often imbalanced [68] and characterized by high dimensionality (commonly known as thick data) [68], and almost always have unknown data distribution [44]. Data of this type cannot easily be analyzed using common parametric statistical tools, but few contributions mentioned nonparametric statistics. As

such, both academics and practitioners should work together to find feasible solutions that can deal with this challenging scenario. Secondly, there is a lack of case studies on the application of big data analytics and machine learning in SC 4.0 [32], confirmed by the results of our analysis: only 15% of studies focused on a practical application of such tools in this field. In other words, there is a need to create links with practice. This aligns with the detected problems: the lack of adequate analysis tools means there are few application examples in the literature. Finally, an instrument is needed to select the most appropriate big data analysis or machine learning techniques. In this way, companies will be able to choose the right instrument to optimize the desired performance.

Finally, we provided some suggestions to deal with typical data flowing in SC 4.0. From the analysis, we saw that data are not always structured, but a high percentage of articles also mentioned unstructured data, which need appropriate methods in order to be properly analyzed. In this case, instruments such as sentiment analysis could be successfully applied. Consequently, we examined these tools. As previously stated, typical manufacturing data are high-dimensional and not normally distributed; therefore, a possible solution for their analysis is nonparametric statistics. As such, we devoted part of this study to examining the nonparametric approach as a possible, viable solution that can be used to perform analyses in this context. We discussed their potential, properties, and advantages.

**Author Contributions:** Conceptualization, E.B., N.B. and R.C.; methodology, E.B., N.B. and R.C.; formal analysis, E.B., N.B. and R.C.; writing—original draft preparation, E.B., N.B. and R.C.; writing—review and editing, L.S.; supervision, L.S. All authors have read and agreed to the published version of the manuscript.

**Funding:** This study was carried out within the MICS (Made in Italy—Circular and Sustainable) Extended Partnership and received funding from the European Union Next-GenerationEU (PIANO NAZIONALE DI RIPRESA E RESILIENZA (PNRR)—MISSIONE 4 COMPONENTE 2, INVESTI-MENTO 1.3—D.D. 1551.11-10-2022, PE00000004). This manuscript reflects only the authors' views and opinions; neither the European Union nor the European Commission can be considered responsible for them.

**Institutional Review Board Statement:** Not applicable.

**Informed Consent Statement:** Not applicable.

**Conflicts of Interest:** The authors declare no conflict of interest.

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
