# Peer review of "Big Data Analytics and Machine Learning in Supply Chain 4.0: A Literature Review"

_stats, doi:10.3390/stats6020038_

Round 1

Reviewer 1 Report

The paper has some good contributions in terms of novelty and contribution. I have the following comments as follows:

1.       The abstract must indicate the significant findings in the last part of it. It should be more scientifically explained, not any informal or casual writing.

2.       The observations are so simple and insignificant. Derive the crucial observations, findings, and insights.

3.       What are the fundamental findings that this paper can be published in this reputed journal? You must explain it.

4.       There are several significant findings in the literature in this direction. Therefore, it is important to obtain the novel findings of this research. There must be a comparative study with the following articles in literature (Optimized radio-frequency identification system for different warehouse shapes; Controllable energy consumption in a sustainable smart manufacturing model considering superior service, flexible demand, and partial outsourcing; A sustainable production-inventory model for a controllable carbon emissions rate under shortages; Joint replenishment strategy for deteriorating multi-item through multi-echelon supply chain model with imperfect production under imprecise and inflationary environment) to show the major contributions and findings.

5.       Keywords should be perfect. The abstract should contain the details of the study and the findings in a very constructive way. A professional proofreading service is highly recommendable for English corrections.

6.       The introduction should be based on the exact research gap, and the literature review should be based on the specific keywords-based review; finally, make an author's contribution table to show the novelty and effectiveness of the study. Show all referenced papers in the table to show the contribution of this study.

7.       Please write the significant findings in conclusions. Do not mention all assumptions which have been indicated within the model.

Good

Author Response

Thanks for your comments and suggestions.

We added the significant findings in abstract and conclusion. We also adjusted the keywords so that the literature review results more aligned with them.

In the introduction we explained the contribution of this study, here we do not focus only the exact research gap since we want to give to the reader a briefly explanation of some basic concepts (Industry 3.0, Industry 3.0, SC 3.0, SC 4.0, big data analytics and machine learning) for our literature research. We think that it is important to introduce and explain some background information with the aim of giving to the reader a better understanding of results and discussion. In addition, as you suggested, we derived crucial observations, findings and insights in the analysis of literature review results. When we explained the second research gap, we integrated our explanation with the contributions you have suggested to us that seem really interesting with our analysis. For what concerns the author’s contribution table, we decided to not show it since we already explained what are the novelty contributions from literature directly by mentioning authors in the paragraph where we presented the results.

You can find our improvements highlighted in blue, please see the attachment.

Reviewer 2 Report

1.     As a reader, I am having difficulty distinguishing between supply chain 4.0 and supply chain 3.0. Could you elaborate on the distinctive features of big data analytics and machine learning in SC 4.0 as compared to SC 3.0?

2.     Similar to the previous question, I am also unsure about the differences between Industry 4.0 and Industry 3.0. Can you please explain the unique aspects of big data analytics and machine learning in Industry 4.0 in contrast to Industry 3.0?

3.     The use of vague terms such as "better," "fewer errors," "more robust," "enhance efficiency," "positive impact," and "more accurate" throughout the article can be confusing for readers. It would be more persuasive if the authors support their statements with specific examples or numerical data. For instance, in line 296, the authors mention that "better forecasting tools than traditional ones allow the profitability of a company to increase." Can you please clarify what constitutes "better forecasting tools" and "traditional ones"?

4.     It would greatly benefit the paper if the authors could revise it to be more concise.

Author Response

Thanks for your comments and suggestions.

In order to better understand the main differences between SC 3.0 and SC 4.0 we added some words to better explain the main features of each one. We did the same for Industry 3.0 and Industry 4.0. We suggested additional papers if readers are interested in further reading.

For what concern the third point, in our analysis we report quantitative information if they are present in literature. Probably the fact that in literature there are limited empirical examples is the reason why we can find few numerical examples. In line 296 we clarify the statement.

You can find our improvement highlighted in red, please see the attachment.

Reviewer 3 Report

The authors searched and analyzed 66 papers from web of science and Scopus to understand the main benefits, challenges and areas of application of big data analysis and machine learning techniques in the area of SC 4.0 and proposed potential statistical methods to solve the current gaps that have been discovered. The paper's coverage of the topic is quite comprehensive and easy to understand.

Some minor improvements are:

  1. The authors found that 35% of the paper analyzed were involved external data. It would be better to describe how the external data were utilized in these papers to improve companies' supply chain.
  2. When proposing the potential solutions of using non-parametric methods and NLP techniques, it would be better to let the audience know what's the current status of methods, i.e among the current literatures, are there discussions around these solutions, how useful they are in improving SC, and how widely they are applied in SC 4.0?

good quality

Author Response

Thanks for your comments and suggestions.

We added a paragraph where we briefly explain in which way external data were utilized to improve companies’ supply chain according to the 35% of papers that mentioned them.

Finally we also explained both for non-parametric and for NLP techniques if there are some recent application of them in the field of SC 4.0 and we discussed them.

You can find the improvements highlighted in orange, please see the attachment.

Round 2

Reviewer 1 Report

The paper can be accepted for publication.

Reviewer 2 Report

all previous issues were addressed, no further comments.